# Waist Circumference as a Risk Factor for Non-Alcoholic Fatty Liver Disease in Older Adults in Guayaquil, Ecuador

**DOI:** 10.3390/geriatrics8020042

**Published:** 2023-04-14

**Authors:** Dayana Cabrera, Jorge Moncayo-Rizzo, Karen Cevallos, Geovanny Alvarado-Villa

**Affiliations:** 1Syneos Health, 1951 NW 7th Avenue, Miami, FL 33136, USA; 2Department of Health Sciences, Universidad de Especialidades Espiritu Santo, Guayaquil 092301, Ecuador

**Keywords:** aged, non-alcoholic fatty liver disease, obesity, waist circumference, elderly, logistic regression

## Abstract

Non-alcoholic liver steatosis is currently considered an epidemic. It involves a broad spectrum of liver diseases, in which older adults constitute a susceptible group. The aim of this study is to identify the role of waist circumference as a risk factor for non-alcoholic fatty liver disease. Methods: A cross-sectional study was carried out in 99 older adults who regularly attended five gerontological centers in the city of Guayaquil, Ecuador. The variables studied were age, gender, independent life, access to complete meals, waist circumference, and NAFLD diagnosed by ultrasound. Results: A significant relationship exists between waist circumference, body mass index, and fat mass percentage. However, only age and waist circumference were significant in the multivariate logistic regression model. Our results suggest that in the presence of waist circumference, body mass index loses its significance and age may be a protective factor due to adipose tissue loss and redistribution. Conclusion: Anthropometric measurements such as waist circumference can be used as complement indicators of NAFLD.

## 1. Introduction

Non-alcoholic fatty liver disease (NAFLD) is characterized by the formation of a cluster of lipid vesicles around the internal structure of hepatocytes. The term was first coined in 1980 by Ludwing J., describing liver diseases that usually develop in alcoholic patients, when these are present in individuals who do not reach a toxic alcoholism level [1]. NAFLD involves a wide variety of liver diseases ranging from steatosis to critical conditions such as cirrhosis [2]. Within its etiology, what is notable is its high frequency in patients with a history of metabolic conditions linked to obesity and type II diabetes [3].

It is estimated that within the general population, between 20% and 30% suffer from NAFLD, and the percentage increases to 70–80% in diabetic individuals [4], which makes it one of the chronic diseases with the greatest impact on a global scale [3]. NAFLD does not have a standard treatment [5] and its characteristics vary according to age, with the elderly being the most vulnerable. With advancing age, liver diseases present greater complications and liver damage than in young individuals [6]. Studies show that diseases such as NAFLD are related to age [7]; however, this is not a significant factor in the degree of steatosis [8], compared to the influence of the type of fatty acid [9].

In Guayaquil, Ecuador, 30 out of every 100 older adults show signs of NAFLD [10], while in Rotterdam, the Netherlands, the number increases to 35 in 100, with increased severity associated with an increase in waist circumference (WC) [11]. Therefore, the aim of this study is to identify the role of waist circumference as a risk factor for non-alcoholic fatty liver disease.

## 2. Materials and Methods

### 2.1. Study Design and Participants

The research was a cross-sectional observational study performed between October and May from 2014 to 2016. This study was approved by the ethics committee of Clinica Kennedy in Guayaquil, Ecuador, on 5 March 2014. The participants were selected by a non-probabilistic convenience sampling method from 5 gerontological centers in Guayaquil, Ecuador. Informed consent was obtained from all the individuals who agreed to participate in the study. The sample was selected based on the following inclusion criteria: age ≥ 65 years, able to attend one of the 5 selected gerontological centers, and able to sign the informed consent. Moreover, subjects with the following characteristics were excluded from the study: consumption of alcohol in quantities > 20 g/day, reporting flatulence at the time of WC measurement, individuals with significant cognitive impairment, and history of liver diseases (Hepatitis B, C, autoimmune disease, biliary disease, cirrhosis).

Once the analysis was completed, each one was given a report of these results with a brief explanation; in addition, the food and transportation expenses previously stipulated in the protocol were covered.

### 2.2. Data Collection

The patients who agreed to participate in the study were scheduled on a specific date for the application of some questionnaires, and the measurement of abdominal circumference and fat mass. Demographic data were collected and the Mini Nutritional Assessment (MNA) [12] was applied to carry out the nutritional screening of the participants, and the percentage of fat mass was obtained with the Tanita scale. For measurement of the waist circumference, a non-elastic plastic measuring tape was used to give a measurement in centimeters of each participant; using two cut-off points as reference [13]: men, greater than 90 cm and women, greater than 80 cm.

The abdominal and pelvic ultrasound was performed by a certified sonographer at the “Fundacion Damas del Honorable Cuerpo Consular” medical center, using a Siemens Sonoline G40 ultrasound. Non-alcoholic liver steatosis was defined by the ultrasonographic as the increase in echogenicity of the liver parenchyma in comparison with the renal cortex and spleen [14,15]. Moreover, the criteria used to define the degree of severity of NAFLD was based on the report by Sherif et al. [16]:Mild (Grade 1): slight, diffuse increase in fine echoes in liver parenchyma with normal visualization of diaphragm and intrahepatic vessel borders;Moderate (Grade 2): moderate, diffuse increase in fine echoes with slightly impaired visualization of intrahepatic vessels and diaphragm;Severe (Grade 3): marked increase in fine echoes with poor or non-visualization of the intrahepatic vessel borders, diaphragm, and posterior right lobe of the liver.

### 2.3. Statistical Analysis

Descriptive statistics were compiled using frequency tables and measures of central tendency. Normality tests were performed, and parametric or non-parametric tests for association were applied as appropriate. For correlation analysis, Pearson or Spearman tests were performed according to the distribution of the data. Moreover, association analysis was performed with a chi-squared test. In addition, a univariate logistic regression and a multivariate logistic regression were run to assess the factors that may influence the development of NAFLD. The dependent variable was the presence or absence of NAFLD. On the other hand, covariates included age, WC, BMI, and the percentage of fat mass.

## 3. Results

### 3.1. Participants

A total of 100 patients who met the inclusion criteria were invited to participate in the study; however, only 99 patients agreed to participate. Table 1 shows the demographic data obtained. The population studied had a mean age of 79 (SD = 4.83) years ranging from 71 to 94. Fifteen percent of the total population were male. Almost all the participants (95%) lived independently. Moreover, 80% of them had three meals a day and 82% were well nourished according to the MNA (mean of 26.3, SD = 2.58, ranging from 19.5 to 30). However, according to the BMI classification and the fat mass classification, the majority of the participants were overweight/obese (72.7% and 68.4%, respectively). The mean BMI was 27.16 (SD = 3.83) ranging from 19.91 to 36.57; and the mean percentage of fat mass was 33.48 (SD = 7.10), ranging from 13.5 to 48.5. Finally, the participants had an average WC of 89.76 (SD = 9.35) cm, ranging from 68 to 119.

### 3.2. Association of Variables with NAFLD

The presence of NAFLD determined by echography was associated with the anthropometric variables. Table 2 shows the relation between these variables. The mean value of the waist circumference, MNA, percentage of fat mass, and BMI was higher in the NAFLD group in comparison to the non-NAFLD group. Moreover, an association was identified between BMI categories and the presence of NAFLD (see Figure 1). Overweight and obesity were more frequent in people with NAFLD (*p* = 0.014). On the other hand, associations between sex, independence, complete meals, and age group were non-statistically significant (*p* > 0.05).

### 3.3. Logistic Regression Explaining the Absence or Presence of NAFLD

Logistic regression was performed to assess the factors that may increase the possibility to develop NAFLD. Two regressions were performed: a univariate logistic regression and a multivariate logistic regression. The first one was performed to assess the odds ratio for each variable as a unique factor. Then, multivariate logistic regression was performed to assess a predictive model. The method used was “intro” with the variable NAFLD (0 = absence; 1 = present) as the dependent variable.

The univariate regression analysis shows that overweight and obesity are direct factors for developing NAFLD (OR: 4.65 (CI: 1.22–17.62) and 7.33 (CI: 1.72–31.34), respectively). Both were statistically significant with *p* values = 0.024 and 0.007, respectively. In addition, the percentage of fat mass is a variable that increases the risk for NAFLD 1.11 times (CI: 1.04–1.2); this was statistically significant (*p* = 0.004).

For the multivariate analysis, the covariates were the anthropometric variables and age recollected from the participants. There were 98 cases analyzed from the total population (1 missing value). The model was statistically significant (X2 = 21.021; Fd = 5; *p* < 0.001); moreover, the model fits well within the data (X2 = 13.375; Fd = 8; *p* = 0.100). This model explained the range from 19.3% to 26.9% of the variance with a total correct prediction of 72.4%, a specificity of 87.9%, and a sensitivity of 40.6%. However, only the age and the waist circumference were statistically significant (*p* = 0.045 and 0.016, respectively). Age presented a negative coefficient (B = −0.115), indicating a decrease in risk with increasing age. On the other hand, the waist circumference presented a positive coefficient (B = 0.092) which indicates an increased risk for NAFLD. Finally, the constant in the model was not statistically significant, which indicates that there are variables that affect the model which is not taken into consideration. Table 3 shows the OR of the evaluated variables.

## 4. Discussion

In the last decade, non-alcoholic liver steatosis has been classified as an epidemic [17], which is influenced by factors such as age, gender, and diabetes, in addition to the trend towards increased obesity in all age groups [18]. The prevalence of the presence of NAFLD is similar to the one reported by Reyes et al. in Guayaquil, Ecuador (32.3% vs. 30%) [10].

Studies report that NAFLD affects men more than women [19]; however, women of post-menopausal age, who constituted 84.8% of the studied population, are equally susceptible due to inflammatory hepatic changes related to estrogen deficiency [20]. In addition, variables such as age are usually a determining factor for the appearance of NAFLD in more than 40% of adults over 60 [21], making population selection highly important. In contrast to our results, we found no association between the presence of NAFLD and age (*p* = 0.198). In spite of that, the multivariate logistic regression showed age as a significant factor that may reduce the probability of NAFLD (OR: 0.892; CI: 0.797–0.998; *p* = 0.045). The literature describes an increase in body fat percentage in proportion to muscular mass due to age-induced sarcopenia [22]. However, our results suggest a decrease in body fat percentage too. This may be due to changes and fat redistribution caused by aging. This was also explored by Kupusinac et al. [23] and Macek et al. [24] who found that people over 60 years of age have less percentage of fat mass in comparison to people 50–60 years of age.

Moreover, some studies have suggested that conditions such as malnutrition have an influence on the development of an aggressive NAFLD [25,26]. However, rapid weight loss has also been reported as beneficial for the improvement in liver steatosis [27], emphasizing diet as the mainstay of disease treatment [28,29]. In this study, no statistically significant relationship was found between the presence of NAFLD and the MNA score (*p* = 0.152).

Another risk factor that stands out in the development of NAFLD is waist circumference [30], which is usually associated with obese patients, though cases of NAFLD have also been observed in non-obese individuals [31]. In the present study, we found a significant relationship between average WC and the presence of NAFLD (*p* = 0.001). Average WC presents a seven-point increase in patients with NAFLD (no-NAFLD = 87.5 vs. NAFLD = 94.5). This is consistent with previous studies indicating the relationship between adiposity indicators and NAFLD [8,32,33]. Moreover, logistic regression showed a significant odds ratio for WC (*p* = 0.018), indicating that people with high WC have a range from 1.017 to 1.197 times the probability to present NAFLD (OR = 1.104).

In addition, some studies present a significant relationship between BMI and the presence of NAFLD [19,32,34,35]. The univariate logistic regression shows a significantly increased risk in patients with overweight and obesity. This is in line with the literature which has established the relationship between obesity, dyslipidemia, metabolic syndrome, and the presence of NAFLD [19,35]. However, even though we found a significant relationship between BMI scores and the presence of NAFLD (26.36 vs. 28.85, *p* = 0.005), the multivariate logistic regression was not statistically significant (*p* = 0.319). This indicates that in the presence of the waist circumference, as an evaluation of abdominal adiposity, the BMI categories lose their significance. This effect has already been explored and reported by several authors [36,37,38,39,40,41].

Finally, the percentage of fat mass was another statistically significant variable when analyzing the relationship with the presence of NAFLD (31.98 vs. 36.55, *p* = 0.009); however, the odds ratio was not significant (*p* = 0.186). In contrast to our results, some studies have investigated the predictor power of the fat mass for NAFLD with statistically significant values [42,43].

The predictive model expressed the importance of abdominal obesity in the development of NAFLD (see Figure 2). This is supported by the biochemical pathophysiology of NAFLD that has been proposed. The literature suggests that some of the mechanisms of NAFLD are low-grade hepatic inflammation and insulin resistance due to free fatty acids released by the visceral adipose tissue [40,44,45]. Furthermore, there are several cytokines named adipokines which are released by the adipose tissue. Adiponectin and leptin are two of the most studied adipokines, which play a major role in the development of NAFLD. Adiponectin is an important hormone that helps with anti-inflammatory effects and helps in tissues’ insulin sensitization. In people with high visceral adipose tissue, the levels of adiponectin are lower. This decrease in adiponectin levels may induce insulin resistance, which affects the hormone-sensitive lipase, resulting in the release of free fatty acid to hepatic circulation [44,46,47].

Moreover, the hyperinsulinemia state due to insulin resistance produces an inhibition of free fatty acid beta-oxidation with the promotion of hepatic fat accumulation as the principal consequence. Moreover, the lower levels of adiponectin may result in the promotion of a pro-inflammatory state. This, in addition to the secretion of pro-inflammatory cytokines, such as IL-6 and TNF-alpha, induced by free fatty acids will increase the insulin resistance state of the adipocytes [40,44,45,46,47]. This results in a positive feedback loop, which implies a constant development of the NAFLD.

Finally, our model suggests that there is a protective effect of age, which is supported by the changes and fat redistribution produced by aging [23,24]. Considering that lower levels of adipose tissue are correlated with higher levels of adiponectin, it can be deduced that age may express a protective effect due to the lower fat mass which results in higher levels of adiponectin, as reported by some authors [48,49,50].

This study has several limitations. The sample is not representative for the study population, which may limit the generalizability of the results. Many variables which may influence the presence of NAFLD were not taken into consideration for this study, such as skeletal muscle mass and comorbidities related to metabolic syndrome including diabetes mellitus, hypertension, and dyslipidemia [11,19,51]. So, results must be interpreted cautiously as many confounding variables were not analyzed. Furthermore, even though liver biopsy has been considered the gold diagnostic method for NAFLD, its routine use is controversial [52], so less invasive techniques such as ultrasound are chosen [53]. Ultrasonography as a diagnostic method has the disadvantage that it is operator dependent [54], not being effective in accurately determining the degrees of steatosis [55]. The technique has shown a high sensitivity and specificity only in the detection of moderate and severe grades [56]. Finally, the MNA score was considered an indirect marker for unhealthy dietary behavior, but there are better tools to assess the dietary pattern of the patients, such as the food frequency questionnaire, food consumption record, and dietary history [57].

## 5. Conclusions

NAFLD represents a public health problem, due to population trends showing an increase in risk factors such as obesity, hypertension, and metabolic syndrome. The use of anthropometric measures such as waist circumference is an indicator of risk for the development of the disease. However, even though many studies show the predictor value of variables such as BMI and fat mass percentage, we found an association with the presence of NAFLD, but no predictor capacity. This implies the use of anthropometric measures as a complement to predict the development of NAFLD.

## Figures and Tables

**Figure 1 geriatrics-08-00042-f001:**
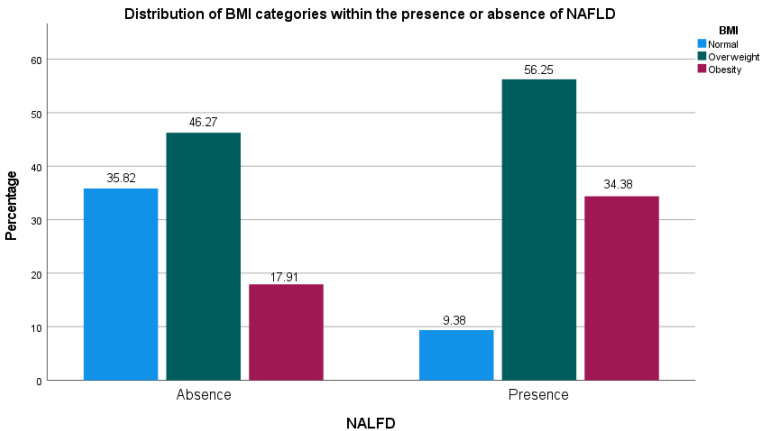
Relationship between BMI categories and the presence or absence of non-alcoholic fatty liver disease.

**Figure 2 geriatrics-08-00042-f002:**
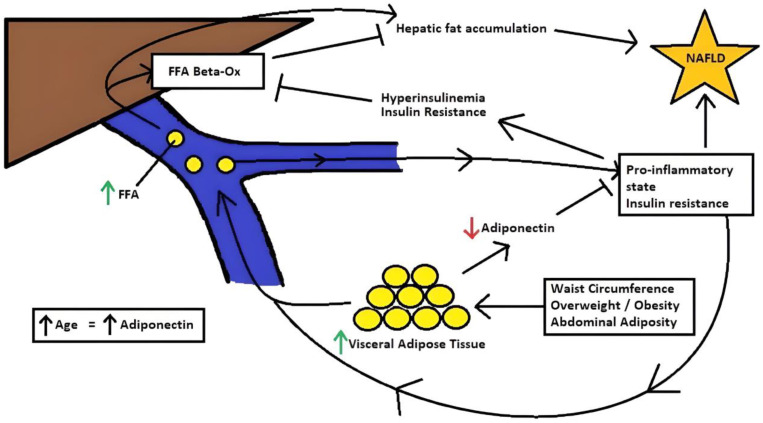
NAFLD mechanism and interaction with anthropometric variables. Increase in waist circumference, along with overweight/obesity and increase in abdominal adiposity will augment the visceral adipose tissue which lead to an incremental release of FFA and a reduction in adiponectin. These predispose to a pro-inflammatory state and insulin resistance, which inhibits FFA beta-oxidation and increases FFA release. As a consequence, there is a hepatic fat accumulation which leads to NAFLD. Finally, due to adipose tissue loss and redistribution, age may lead to a slight incremental increase in adiponectin, which induces an anti-inflammatory state and insulin sensitization. FFA: free fatty acids. Beta-Ox: beta-oxidation. NAFLD: non-alcoholic fatty liver disease.

**Table 1 geriatrics-08-00042-t001:** Characteristics of the participants.

		*n* = 99	Percentage (%)
Sex	Man	15	15.2%
	Woman	84	84.8%
Age	71–75	20	20.2%
	76–80	45	45.5%
	81–85	19	19.2%
	86–90	13	13.1%
	91–95	2	2.0%
NAFLD	Normal	67	67.7%
	Grade I	29	29.3%
	Grade II	3	3.0%
Lives independently	No	4	4.1%
	Yes	94	95.9%
Complete meals	Two meals	19	19.2%
	Three meals	80	80.8%
MNA Classification	Well-nourished	82	82.8%
	Risk of malnutrition	17	17.2%
BMI Classification	Normal	27	27.3%
	Overweight	49	49.5%
	Obesity	23	23.2%
Fat Mass	Athlete	2	2.0%
	Fitness	6	6.1%
	Acceptable	23	23.5%
	Obesity	67	68.4%

MNA: Mini Nutritional Assessment; BMI: body mass index; NAFLD: non-alcoholic fatty liver disease.

**Table 2 geriatrics-08-00042-t002:** Association of variables with NAFLD presence.

	Normal	NAFLD	*p* Value
	Mean (SD)	Mean (SD)	
Age	79.75 (5.12)	77.69 (3.86)	0.198
Waist circumference (cm)	87.50 (8.32)	94.50 (9.74)	0.001 *
MNA	26.00 (2.54)	26.94 (2.64)	0.152
BMI	26.36 (3.58)	28.85 (3.83)	0.005 *
Percentage of fat mass	31.98 (7.27)	36.55 (5.67)	0.009 *

MNA: Mini Nutritional Assessment; BMI: body mass index; NAFLD: non-alcoholic fatty liver disease. * *p* value < 0.01.

**Table 3 geriatrics-08-00042-t003:** Multivariate logistic regression which explained the presence of NAFLD.

Variables	B	Sig.	OR	OR 95% CI	
				Inferior	Superior
Age	−0.115	0.045 *	0.892	0.797	0.998
Waist circumference (cm)	0.092	0.016 *	1.097	1.017	1.183
Overweight	0.473	0.565	1.605	0.321	8.036
Obesity	−0.657	0.586	0.518	0.049	5.498
Percentage of fat mass	0.07	0.186	1.073	0.967	1.191
Constant	−2.641	0.614	0.071		

* *p* value < 0.05.

## Data Availability

The data presented in this study are available on request from the corresponding author.

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
