# Peer review of "Waist Circumference as a Risk Factor for Non-Alcoholic Fatty Liver Disease in Older Adults in Guayaquil, Ecuador"

_geriatrics, 2023, doi:10.3390/geriatrics8020042_

Round 1
Reviewer 1 Report
The authors conducted a multi-center, cross-sectional study to evaluate the association of waist circumference and NASH. The topic is of great public health importance. The conclusions align with the results presented in the manuscript. Suggestions for improving the manuscript:
1) In the methods section, please move lines 73-75 in section 2.2 (consent, ethics committee approval to section 2.1.
2) In methods 2.1; is there data on the response rate i.e. how many were approached vs how many agreed to participate? If yes, please mention.
3) In section 2.2, lines 67-68: please define the grading criteria used to ascertain NASH i.e. how were the different grades of NASH defined?
4) In section 2.3, (i) line 78: please specify which parametric and non-parametric tests were used. (ii) Line 80- please specify which covariates were used to build the logistic regression model.
5) Please spell out full form of all abbreviations in the footnote for all tables in the manuscript.
6) In the discussion section, please add a figure (concept mode) displaying the potential biological mechanisms explaining the association of waist circumference with NASH.
7) In the limitations para (lines 152-158), please discuss more limitations- e.g. selection bias, recall bias, & potential misclassification of predictor variables.
Reviewer 2 Report
The authors state the aim of the study as – “To identify the relationship between the degree of non-alcoholic liver steatosis and waist circumference”.
There appears to be a disconnect between the stated aim of the study and its main findings. The choice of “presence or absence of NAFLD” as the dependent variable seeks to demonstrate the association between the presence of NAFLD and waist circumference /various risk factors. If that is not what the authors intended, then it makes sense to use “grades of steatosis as defined by ultrasound” as your primary outcome of interest as it aligns well with your chosen aim.
Clearly specify how a case of NAFLD was defined
Clearly specify what grading system was employed in grading the severity of steatosis as defined by ultrasound and define what each grade signify.
Did authors collect information on other factors including other components of metabolic syndrome (eg. systemic hypertension, T2DM) which are likely to confound the association between the presence of Non-alcoholic fatty liver disease and waist circumference?
Clearly state what type of study design was employed in specifically in the method section and also the duration of the study is somewhat is clear ( “This study was conducted in the period from October to May in 2014”)
Specify how you obtained a sample of participants your target population and how participants were subsequently enrolled into the study. Is the sample representative of the general population and are your results generalizable?
Could analyze BMI as a categorical variable specifying categories as normal BMI, overweight, and obesity. I think it is worth exploring the association between DMI as a categorical variable and the presence of non-alcoholic fat liver disease.
Consider exploring alternative proxy measure of unhealthy dietary patterns as the Mimi Nutritional Assessment score may not well capture your intended assessment of unhealthy dietary behavior.
Round 2
Reviewer 1 Report
Thank you for submitting a revised manuscript. Comments from previous round of review have been addressed adequately. I have no additional comments.